# Nutrition Labeling Usage Influences Blood Markers in Body-Size Self-Conscious Individuals: The Korean National Health and Nutrition Examination Survey (KNHANES) 2013–2018

**DOI:** 10.3390/ijerph17165769

**Published:** 2020-08-10

**Authors:** Su Yeon Kye, Kyu-Tae Han, Sung Hoon Jeong, Jin Young Choi

**Affiliations:** Division of Cancer Control & Policy, National Cancer Control Institute, National Cancer Center, 323 Ilsan-ro Ilsandong-gu Goyang, Goyang 10408, Korea; sykye@ncc.re.kr (S.Y.K.); kthan@ncc.re.kr (K.-T.H.); ko0743@naver.com (S.H.J.)

**Keywords:** nutrition labeling, high-density lipoprotein cholesterol, triglycerides, body-size perception

## Abstract

This study analyzed the effects of nutrition labeling and examined whether nutrition labeling usage influences the levels of blood markers, such as high-density lipoprotein cholesterol (HDL-C) and triglyceride (TG) in body-size self-conscious individuals. The dependent variables were HDL-C and TG; the independent variables were the respondents’ awareness of nutrition labeling use, sociodemographic factors, perceived health status, stress, lifestyle, frequency of eating out, family history of hyperlipidemia, survey year, body mass index, total energy intake, and cholesterol levels. Body-size perception was assessed by matching body mass index with subjective body-shape recognition using data from the Korean National Health and Nutrition Examination Survey (2013–2018). Differences were observed in HDL-C and TG levels according to nutrition labeling usage and body-image perception. The group that recognized body image correctly showed high HDL-C and low TG levels when they actively used nutrition labeling, whereas the group that recognized body image incorrectly showed no significant changes in HDL-C and TG levels even when actively using nutrition labeling. The standard nutritional information, which does not consider individual body-size perceptions, has a restrictive effect. Policies should be developed towards tailored intervention strategies considering individual body-size perception.

## 1. Introduction

As communication and information technology has developed, and people’s interest in health has increased, our contemporary world is experiencing an explosive growth in health information [1]. Evidence-based accurate health information improves health-related knowledge and has a positive impact on the enhancement of individual health behaviors, such as increase in physical activity, intake of vegetables and fruits, and regular cancer screening [2,3]. However, misuse of limited health information could impede an individuals’ awareness regarding diseases and compliance with medication [4,5].

Information regarding diet accounts for a significant proportion of all health information available on various media platforms [6]. Many countries have adopted nutrition labeling on food products as a policy for empowering individuals to make informed decisions and opt for healthy food choices, thereby enabling them the right to make decisions regarding their health while purchasing food [7,8]. In 1995, South Korea made nutrition labeling mandatory. Labeling also prevents consumers from succumbing to misleading advertisements on the nutritious value of food products, as accurate nutritional information is provided. Previous studies have revealed that nutrition labeling enabled people to make better food choices, improved the quality of food products being consumed, and positively affected nutrition-related biomarkers, such as high-density lipoprotein cholesterol (HDL-C), triglycerides (TG), and insulin resistance [9,10,11,12,13]. However, it is challenging to find studies that have categorized individuals according to varied personal characteristics in the existing literature, which mainly comprises studies conducted on general characteristics or involving individuals with specific diseases [9,10,11,12,13].

To practice a healthy lifestyle, it is important to cultivate habits that increase health-related knowledge and beliefs, improve attitude, and enhance self-efficacy [14]. Tailoring, which can be defined as any of the several methods applied for creating communications that are customized, is generally considered to be a useful method for increasing the effectiveness of health interventions, through the delivery of tailored print, as well as telemedicine or eHealth applications [15,16]. Body image could be an important personal factor for tailoring. Development of a healthy perception of one’s body image is the first step in preventing and managing obesity and is essential for forming healthy eating habits as well as for healthy weight control [17,18]. Body image refers to how one perceives, feels, and thinks about one’s body [19]. Distorted body images cause psychological problems, such as over-exercising, eating disorders, decline in self-esteem, depression, and physical problems, such as malnutrition, osteoporosis, digestive problems, and cardiovascular diseases [20,21]. Previous studies have focused on how intervention strategies affect body image, but little attention has been paid towards research using body image as a resource to develop intervention strategies for healthy eating habits [22,23].

Dyslipidemia is a state of abnormal amounts of lipids in the blood and is characterized by conditions such as decreased HDL-C (≤40 mg/dL) and hypertriglyceridemia (≥150 mg/dL) [24]. Although not necessarily harmful itself, the condition is a major risk factor for various cardiovascular diseases. Plasma concentrations of HDL-C have shown a strong inverse relationship with the risk of developing coronary artery disease and cancer [25]. Conversely, a low concentration of HDL-C is associated with obesity [25]. The purpose of this study was to analyze the effects of active use of nutrition labeling and to examine whether body image can be an important criterion for identifying target groups, while developing strategies for cultivating healthy habits. Therefore, we examined whether there were differences in the levels of HDL-C and TG according to the usage of nutrition labeling in Korean people who have never been diagnosed with hypertension, diabetes, or dyslipidemia, which are chronic diseases related to diet. We also examined whether nutrition labeling usage influences the levels of HDL-C and TG in body-size self-conscious individuals.

## 2. Materials and Methods

### 2.1. Participants

This study used data from the 2013–2018 Korean National Health and Nutrition Examination Survey (KNHANES) [26]. These cross-sectional questionnaires have been administered annually since 1998 by the Korea Centers for Disease Control and Prevention to assess the health and nutritional status of the general Korean population. The KNHANES is composed of three parts: Health Interview Survey, Health Examination, and Nutrition Survey. The data for each year were collected via the national representative survey using a multistage cluster-sampling design. After providing informed consent, participants completed an extensive interview and underwent assessment at a mobile examination center by trained interviewers and medical staff. The nutrition survey was conducted at the participants’ homes 1 week after the health interview. The KNHANES is approved by the institutional review board of the Korea Centers for Disease Control and Prevention. Detailed information about KNHANES has been provided elsewhere [27]. Between 2013 and 2018 a total of 47,217 participants were enrolled. Respondents who aged less than 19 were excluded from these analyses (*n* = 6453). We also excluded respondents diagnosed with hypertension, diabetes, or dyslipidemia from the study (*n* = 8065/*n* = 3163/*n* = 5621), and those with missing values for the diagnosis of the diseases (*n* = 558). Additionally, individuals not tested for dyslipidemia indicators such as HDL-C and TG or those with no response for independent variables were excluded (*n* = 5922). A total of 16,619 participants were deemed eligible for this study (Figure 1).

### 2.2. Measurements

#### 2.2.1. Outcome Variables

##### Levels of HDL-C and TG

To identify the effects of active use of nutrition information according to one’s body-image perception, the outcome variables analyzed in this study included two indicators of dyslipidemia: HDL-C and TG. KNHANES collected data through household member confirmation surveys, health questionnaire surveys, examination surveys, and nutrition surveys. The examination surveys consisted of physical measurement, blood pressure and pulse measurement, blood and urine test, oral examination, lung function test, eye test, and grip test. The tests were performed on the day of the survey. The blood test conducted was a fasting blood test (minimum 8 h and recommended 12 h after eating), performed by a trained nurse in the mobile examination vehicle. These indicators were considered continuous variables in the analysis.

#### 2.2.2. Independent Variables

The primary independent variable was the participants’ active use of nutrition labeling, which was categorized as follows: (1) checks nutrition labeling and makes label-dependent purchase decisions and (2) checks nutrition labeling but does not make label-dependent purchase decisions or is aware of nutrition labeling but does not check the labels when making food purchase decisions or is unaware of nutrition labeling (i.e., not actively used).

Other independent variables included sex, age, educational level, economic status, household income, marital status, residential area, perceived health status, stress awareness, alcohol intake, smoking status, aerobic exercise habits, walking for more than 10 min, frequency of eating out, family history of hyperlipidemia, survey year, body mass index (BMI), total energy intake, and cholesterol levels [28,29,30,31]. The participants were grouped according to 10-year age increments, and those aged 70 years or older were grouped into one category. Education level was categorized as follows: high school or below, bachelor’s degree, and master’s degree and above. Perceived health status and stress awareness were categorized as good/high or bad/low. Aerobic exercise habits were based on the amount of aerobic exercise per week, with 150 min of exercise as the cutoff. Total energy intake was calculated based on food intake on the day before the survey (24-h recall method).

#### 2.2.3. Body-Size Perception

To identify the participant’s perception of body image, a body-size perception variable was created by combining BMI and subjective body-shape recognition. BMI was defined as underweight, normal, and obese based on obesity criteria in South Korea (<18.5, 18.5–25, and >25, respectively) [32]. Subjective body-shape recognition was the participants’ perception of his/her own body size, which was divided into three categories: thin, normal, and fat. Body-size perception was categorized into three groups: misperception (over), healthy perception, and misperception (under). When subjective body-shape recognition was exaggerated in comparison with the actual body shape, body-size perception was defined as misperception (over), whereas it was defined as misperception (under) when subjective body-shape recognition was understated. When participants perceived their actual body shape correctly, body-size perception was defined as healthy perception.

### 2.3. Statistical Analysis

We first examined the frequencies and percentages of categorical variables and means and standard deviations of continuous variables. Next, we performed an analysis of variance to determine the association of HDL-C and TG levels with the categorical variables. Multiple linear regression analysis was conducted to investigate the relationship between active use of nutrition labeling and HDL-C and TG levels, adjusting for other independent variables. In addition, we carried out subgroup multiple linear regression analysis by body-size perception to examine differences in nutrition labeling awareness and their effect on the HDL-C and TG levels of the participants. Considering the distribution of outcome variables, we log-transformed them; the results of regression analysis could be interpreted as percent changes after exponentiating the coefficients and multiplying by 100. Sampling weights assigned to each participant were applied in the analysis to generalize the sampled data. All statistical analyses were performed using SAS version 9.4 (SAS Institute Inc., Cary, NC).

## 3. Results

Male individuals comprised 49.45% of all respondents, and 64.38% of the participants were married. Of the respondents, 87.43% perceived themselves as being healthy and 7.01% had a family history of hyperlipidemia. Approximately 26% of the participants checked nutrition labeling and actively used it while making decisions about purchasing food, whereas 73% checked nutrition labeling but did not make label-dependent purchase decisions or were aware of nutrition labeling but did not check the labels while making food purchase decisions or were unaware of nutrition labeling. Regarding body-size perception, 18.53% overestimated, 15.11% underestimated, and 66.36% appropriately recognized their body shape (Table 1).

The average BMI in this study was 23.45 kg/m^2^. The value was higher in males than females. The respondents had a total energy intake of about 2150 kcal in a day, and females had a lower intake of calories in a day. The average total cholesterol was 191.20 mg, and it was similar in both sexes (Table 2).

Table 3 shows results of analysis of variance to examine associations between the independent and outcome variables to compare the association of HDL-C and TG levels with the categorical variables. The average HDL-C and TG levels were 52.40 and 126.27 mg/dL, respectively. There was a nonsignificant relationship between active use of nutrition labeling and outcome variables (*p* = 0.2282; *p* = 0.0653). HDL-C levels were significantly higher in respondents with female sex (*p* < 0.0001), younger age (*p* = 0.0008), higher household income (*p* = 0.0319), single status (*p* < 0.0001), better perceived health status (*p* = 0.0004), low stress awareness (*p* = 0.0138), higher alcohol intake (*p* < 0.0001), non-smoking status (*p* < 0.0001), aerobic exercise habits (*p* = 0.0007), or frequent walking for more than 10 min (*p* < 0.0001). TG levels were significantly lower in participants with female sex (*p* < 0.0001), younger age (*p* < 0.0001), bachelor’s degree (*p* = 0.0002), single status (*p* = 0.0087), low stress awareness (*p* = 0.0133), lower alcohol intake (*p* < 0.0001), nonsmoking status (*p* < 0.0001), aerobic exercise habits (*p* = 0.0003), frequent walking for more than 10 min (*p* = 0.0125), or no family history of hyperlipidemia (*p* = 0.0141).

Table 4 shows results of multiple linear regression to examine the association between active use of nutrition labeling and outcome variables. Individuals who used nutrition labeling actively had significantly higher HDL-C (1.18% increases) and lower TG levels (3.18% decreases) (*β* = 0.012, *p* = 0.0079; *β* = −0.032, *p* = 0.0029). Male or older individuals generally had higher risk levels associated with the two indicators, whereas individuals with healthy behaviors had lower risk levels associated with the two indicators, with the exception of alcohol intake.

We also conducted a subgroup multiple linear regression analysis to investigate relationships between active use of nutrition labeling and outcome variables according to body-size perception. In the group that comprised participants who had healthy awareness of their body size, the participants who actively used nutrition labeling had higher HDL-C (1.36% increases) and lower TG levels (−3.24% decreases) than those who did not actively use nutrition labeling (*β* = 0.013, *p* = 0.0100; *β* = −0.033, *p* = 0.0105). In contrast, in the group that comprised participants who did not have a healthy awareness of their body size, there were not significant differences in HDL-C and TG levels according to the active use of nutrition labeling (*β* = 0.010, *p* = 0.4438; *β* = −0.036, *p* = 0.2457; *β* = 0.010, *p* = 0.3019; *β* −0.031, *p* = 0.1430) (Table 5).

## 4. Discussion

This study aimed to analyze the influence of nutrition labeling on HDL-C and TG levels and to examine the possibility of adopting body-size perception as a criterion for identifying target groups when developing strategies for cultivating health skills. The findings revealed that there were differences in HDL-C and TG levels according to the usage of nutrition labeling by the participants, and the effects of active use of nutrition labeling on HDL-C and TG levels differed on the basis of a participant’s perception of their body image.

Respondents who checked nutrition labeling and made label-dependent purchase decisions were more likely to show a higher level of HDL-C and lower level of TG than were those who did not use nutrition labeling actively. This finding is in line with those of previous studies, indicating that the habits of reading nutrition information and applying it in real life could yield healthy outcomes, thereby emphasizing the importance of dissemination and sharing of health information [7,12]. Checking and using nutrition labeling had a large effect on the management of HDL-C levels in cancer survivors [13]. Active use of nutrition labeling could have positive effects in reducing insulin resistance and preventing diabetes mellitus [9]. Individuals who used nutrition labeling to make decisions regarding food purchases had a higher level of HDL-C and lower levels of TG [11]. As a result of active use of nutrition labeling, healthy dietary decisions were made by the participants while purchasing food, and therefore, it appears that nutrition-related biomarkers such as HDL-C and TG could be influenced by nutrition labeling. In other words, the introduction of a nutrition labeling system could be beneficial in adopting health behavior, reducing obesity, and improving health outcomes [33].

Our subgroup analysis regarding body-size perception revealed interesting findings. The results of regression analysis after adjusting for almost all variables, revealed that the group of participants that recognized their body image correctly showed high HDL-C and low TG levels when they actively used nutrition labeling, but the other group of participants, who did not correctly recognize their body size and overestimated or underestimated their body shape, did not show significant changes in HDL-C and TG levels even if they actively use nutrition labeling. This shows that nutrition information does not play an appropriate role if people do not recognize their body size correctly, indicating that exact recognition of individual body image is important for healthy eating. A significant predictor of disturbed eating attitudes and behaviors was body-image dissatisfaction, indicating that a negative body-image perception was one of the key factors contributing to eating disorders [34,35]. In addition, standard nutritional information that does not fit one’s body-size perception could be inefficient, and when not tailored to individuals, this information could not be expected to yield maximum benefits. A randomized controlled trial in a clinical setting reported that tailored advice from general practitioners was effective in improving all individual health behaviors, including a healthy diet [36]. A meta-analysis also showed that computer-tailored interventions increased the absolute rate of fruit and vegetable consumption from 27% to 37%, a meaningful change that is highly recommended to prevent and control obesity and multiple chronic diseases [37]. In addition, a randomized controlled trial revealed that a 6-month tailored intervention delivered via a mobile phone application improved fruit and vegetable intake [38]. The findings of this study indicate the difference in the effect of nutrition labeling usage according to body-size perception and shows that body images could be an important criterion for developing intervention strategies to improve health skills for a healthy diet. Hence, it appears that providing tailored nutrition information according to body-size perception could have maximum benefits.

### Strengths and Limitations

This study has several strengths and limitations. The strengths include: first, we used nationwide sampling data over a 6-year duration, thereby providing useful results that could help in the establishment of long-term health policies at the national level. Second, our study included clinical outcome variables, such as the results of blood tests, making our findings more valuable than those of other studies using different types of data. The limitations include: first, owing to a cross-sectional study design rather than a longitudinal one, there are some concerns about causal relationships between nutrition labeling awareness and outcome variables. Second, active use of nutrition labeling, perceived health status, and stress awareness were assessed using a self-reported survey, which raises issues related to recall bias of the participants. Despite these limitations, to the best of our knowledge, this study is the first study to have proposed an intervention strategy for positively improving the dietary behaviors of individuals by considering body-size perception. This would not only enhance the usefulness of nutrition labeling but also improve the effect of nutrition information.

## 5. Conclusions

Appropriate perception of body size and motivation based on body size help in obtaining optimal results from health behaviors. With the enormous amount of indiscriminate health information, one is likely to be deceived about what applies to them, so it is necessary to select and apply health information suitable to one’s own condition to obtain good results. Healthcare professionals and health communication experts should try to make participants in the target group assess their body-image perception and provide appropriate health information on the basis of their perception, and it will be helpful to use tailored intervention strategies according to individual body-size perception for efficient allocation of resources.

## Figures and Tables

**Figure 1 ijerph-17-05769-f001:**
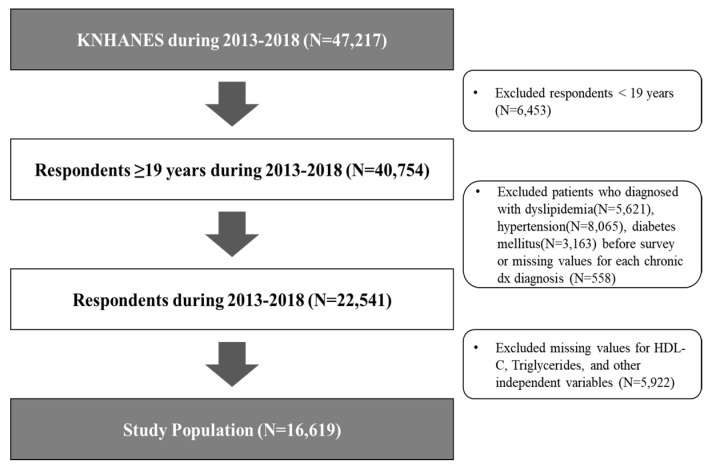
Flowchart of participants included in the main analysis.

**Table 1 ijerph-17-05769-t001:** General characteristics of the study population: categorical variables.

Variables	Study Population		Male		Female	
	N	%	N	%	N	%
Active use of nutrition labeling						
Check and make dependent purchase decisions	4308	26.04	1052	16.81	3256	35.06
Not actively using	12,311	73.96	6075	83.19	6236	64.94
Sex						
Male	7127	49.45	7127	100.00		
Female	9492	50.55			9492	100.00
Age (years)						
<20	264	1.94	140	2.13	124	1.76
<30	2735	22.67	1211	24.13	1524	21.24
<40	4058	25.42	1591	24.45	2467	26.36
<50	4024	24.02	1505	21.85	2519	26.15
<60	2942	16.34	1233	16.40	1709	16.29
<70	1669	6.52	874	7.28	795	5.77
70+	927	3.09	573	3.77	354	2.42
Educational level						
High school or below	7294	39.25	2958	35.67	4336	42.75
Bachelor’s degree	8325	54.58	3668	57.21	4657	52.01
Master’s degree and above	1000	6.17	501	7.12	499	5.24
Economic status						
Employed	5556	31.87	1587	21.38	3969	42.12
Unemployed	11,063	68.13	5540	78.62	5523	57.88
Household income						
Low	3844	24.05	1638	23.76	2206	24.33
Mid-low	4200	25.43	1819	25.95	2381	24.91
Mid-high	4282	25.48	1848	25.42	2434	25.54
High	4293	25.04	1822	24.87	2471	25.21
Marital status						
Single	3776	29.69	1941	35.32	1835	24.19
Separated/divorced/bereavement	1261	5.93	346	3.86	915	7.95
Married	11,582	64.38	4840	60.83	6742	67.86
Residence Area						
Metropolitan	9960	60.29	4178	59.11	5782	61.45
Others	6659	39.71	2949	40.89	3710	38.55
Perceived health status						
Good	14,484	87.43	6341	89.09	8143	85.80
Bad	2135	12.58	786	10.91	1349	14.20
Stress awareness						
High	4524	28.36	1762	26.24	2762	30.43
Low	12,095	71.64	5365	73.76	6730	69.57
Alcohol intake						
Less than 1 time per month	6176	34.18	1739	23.41	4437	44.72
1–3 times per week	9402	59.6	4605	66.83	4797	52.53
More than 4 times per week	1041	6.21	783	9.76	258	2.75
Smoking status						
Smoker	3305	23.16	2743	40.03	562	6.64
Ex-smoker	3256	19.77	2607	32.76	649	7.06
Non-smoker	10,058	57.08	1777	27.21	8281	86.30
Aerobic exercise habits						
Yes	7539	48.34	3432	48.56	4107	45.31
No	9080	51.66	3695	51.44	5385	54.69
Walking for more than 10 min						
Few	2515	14.27	1157	15.24	1358	13.32
1–4 days per week	5133	30.18	2109	28.97	3024	31.36
5–7 days per week	8971	55.55	3861	55.80	5110	55.31
Frequency of eat out						
More than 4 times per week	9762	63.83	5111	76.38	4841	48.45
Less than 3 times per week	6857	36.17	2016	23.62	4651	51.55
Family history of hyperlipidemia						
Yes	1158	7.01	368	5.63	790	8.36
No	15,461	92.99	6759	94.37	8702	91.64
Year						
2013	2715	15.94	1173	16.18	1542	15.69
2014	2491	15.57	1034	15.53	1457	15.60
2015	2564	16.52	1126	16.54	1438	16.49
2016	2854	16.55	1177	16.17	1677	16.94
2017	2917	17.17	1263	16.94	1654	17.39
2018	3078	18.25	1354	18.64	1724	17.88
Body-size perception						
Misperception (over)	3051	18.53	1822	24.30	824	8.07
Misperception (under)	2646	15.11	613	9.11	2438	26.53
Healthy perception	10,922	66.36	4692	66.60	6230	65.40
Total	16,619	100.00	7127	100.00	9492	100.00

**Table 2 ijerph-17-05769-t002:** General characteristics of the study population: continuous variables.

Variables	Study Population		Male		Female	
	Mean	SD	Mean	SD	Mean	SD
BMI (kg/m^2^)	23.45	0.03	24.28	0.05	22.64	0.05
Total energy intake (kcal)	2145.85	9.88	2508.22	15.48	1791.34	8.76
Cholesterol (mg)	191.20	0.33	192.14	0.47	190.29	0.41

**Table 3 ijerph-17-05769-t003:** Association between active use of nutrition labelling and HDL-C and TG levels.

Variables	N	HDL-C (mg/dL)			TG (mg/dL)		
		Mean	SD	*p*-Value	Mean	SD	*p*-Value
Active use of nutrition labelling							
Check and make dependent purchase decisions	4308	54.42	0.23	0.2282	110.61	1.46	0.0653
Not actively using	12,311	51.69	0.14		131.78	1.24	
Sex							
Male	7127	48.10	0.15	<0.0001	153.40	1.75	<0.0001
Female	9492	56.61	0.16		99.72	0.81	
Age (years)							
<20	264	54.61	0.77	0.0008	91.28	3.78	<0.0001
<30	2735	54.01	0.27		103.38	1.64	
<40	4058	52.85	0.21		127.28	2.05	
<50	4024	52.10	0.23		137.11	2.40	
<60	2942	51.42	0.27		141.76	2.27	
<70	1669	49.81	0.34		134.75	2.56	
70+	927	48.59	0.45		123.65	2.92	
Educational level							
High school or below	7294	51.68	0.17	0.1423	135.84	1.60	0.0002
Bachelor’s degree	8325	52.99	0.17		119.16	1.32	
Master’s degree and above	1000	51.84	0.47		128.23	3.88	
Economic status							
Employed	5556	53.73	0.21	0.9016	114.36	1.38	0.0586
Unemployed	11,063	51.78	0.14		131.84	1.32	
Household income							
Low	3844	51.92	0.23	0.0319	130.64	2.10	0.0847
Mid-low	4200	52.04	0.23		129.12	2.05	
Mid-high	4282	52.42	0.22		125.16	1.81	
High	4293	53.21	0.24		120.30	1.83	
Marital status							
Single	3776	53.54	0.24	<0.0001	113.39	1.87	0.0087
Separated/divorced/bereavement	1261	51.87	0.41		140.73	4.18	
Married	11,582	51.93	0.13		130.87	1.15	
Residence Area							
Metropolitan	9960	52.64	0.15	0.5638	123.02	1.24	0.1663
Others	6659	52.05	0.20		131.20	1.69	
Perceived health status							
Good	14,484	52.55	0.13	0.0004	125.39	1.05	0.146
Bad	2135	51.42	0.30		132.35	2.88	
Stress awareness							
High	4524	52.32	0.22	0.0138	129.51	2.01	0.0133
Low	12,095	52.44	0.14		124.98	1.12	
Alcohol intake							
Less than 1 time per month	6176	51.52	0.18	<0.0001	113.95	1.22	<0.0001
1–3 times per week	9402	52.68	0.16		127.61	1.32	
More than 4 times per week	1041	54.60	0.51		181.14	6.20	
Smoking status							
Smoker	3305	48.52	0.25	<0.0001	169.75	2.94	<0.0001
Ex-smoker	3256	50.46	0.24		138.83	2.21	
Non-smoker	10,058	54.65	0.15		104.27	0.88	
Aerobic exercise habits							
Yes	7539	52.91	0.17	0.0007	122.68	1.53	0.0003
No	9080	51.93	0.16		129.62	1.33	
Walking for more than 10 min							
Few	2515	50.99	0.26	<0.0001	131.95	2.48	0.0125
1–4 days per week	5133	51.89	0.21		131.75	1.89	
5–7 days per week	8971	53.05	0.16		121.83	1.31	
Frequency of eating out							
More than 4 times per week	9762	52.07	0.15	0.1223	128.56	1.29	0.1362
Less than 3 times per week	6857	53.00	0.19		122.22	1.45	
Family history for hyperlipidemia							
Yes	1158	54.05	0.48	0.1396	128.26	4.01	0.0141
No	15,461	52.28	0.12		126.12	1.03	
Total	16,619	52.40	0.12		126.27	1.01	

HDL-C: high-density lipoprotein cholesterol; TG: triglycerides.

**Table 4 ijerph-17-05769-t004:** Results of multiple regression analysis for the association between active use of nutrition labelling and outcome variables.

Variables	HDL-C (mg/dL)			TG (mg/dL)		
	β	SE	*p*-Value	β	SE	*p*-Value
Active use of nutrition labelling						
Check and make dependent purchase decisions	0.012	0.004	0.0079	−0.032	0.011	0.0029
Not actively using	Ref	-	-	Ref	-	-
Sex						
Male	−0.150	0.005	<0.0001	0.236	0.012	<0.0001
Female	Ref	-	-	Ref	-	-
Age (years)						
<20	0.105	0.017	<0.0001	−0.097	0.044	0.0287
<30	0.094	0.012	<0.0001	−0.090	0.030	0.0031
<40	0.074	0.011	<0.0001	−0.016	0.026	0.5315
<50	0.048	0.010	<0.0001	0.040	0.024	0.1006
<60	0.032	0.010	0.0021	0.046	0.024	0.0564
<70	0.017	0.011	0.1121	0.021	0.026	0.4191
70+	Ref	-	-	Ref	-	-
Educational level						
High school or below	0.013	0.008	0.1082	−0.001	0.020	0.9793
Bachelor’s degree	0.009	0.008	0.2213	−0.010	0.019	0.5880
Master’s degree and above	Ref	-	-	Ref	-	-
Economic status						
Employed	−0.001	0.004	0.7727	0.028	0.010	0.0033
Unemployed	Ref	-	-	Ref	-	-
Household income						
Low	−0.009	0.006	0.1226	0.031	0.014	0.0243
Mid-low	−0.009	0.005	0.1165	0.024	0.013	0.0699
Mid-high	−0.005	0.005	0.3555	0.023	0.013	0.0638
High	Ref	-	-	Ref	-	-
Marital status						
Single	0.026	0.007	<0.0001	−0.002	0.017	0.9177
Separated/divorced/bereavement	−0.005	0.008	0.5187	0.048	0.019	0.0122
Married	Ref	-	-	Ref	-	-
Residence Area						
Metropolitan	−0.001	0.004	0.8193	−0.004	0.011	0.7401
Others	Ref	-	-	Ref	-	-
Perceived health status						
Good	0.009	0.005	0.1091	−0.013	0.014	0.3533
Bad	Ref	-	-	Ref	-	-
Stress awareness						
High	−0.010	0.004	0.0153	0.020	0.010	0.0499
Low	Ref	-	-	Ref	-	-
Alcohol intake						
Less than 1 time per month	−0.142	0.009	<0.0001	−0.142	0.024	<0.0001
1–3 times per week	−0.089	0.008	<0.0001	−0.129	0.023	<0.0001
More than 4 times per week	Ref	-	-	Ref	-	-
Smoking status						
Smoker	−0.040	0.006	<0.0001	0.190	0.014	<0.0001
Ex-smoker	0.013	0.006	0.0162	0.034	0.013	0.0095
Non-smoker	Ref	-	-	Ref	-	-
Aerobic exercise habits						
Yes	Ref	-	-	Ref	-	-
No	−0.017	0.004	<0.0001	0.050	0.010	<0.0001
Walking for more than 10 min						
Few	−0.014	0.006	0.0124	−0.009	0.014	0.5196
1–4 days per week	−0.015	0.004	0.0005	0.021	0.011	0.0542
5–7 days per week	Ref	-	-	Ref	-	-
Frequency of eating out						
More than 4 times per Week	−0.003	0.004	0.5391	0.011	0.010	0.2717
Less than 3 times per week	Ref	-	-	Ref	-	-
Family history for hyperlipidemia						
Yes	−0.006	0.008	0.4780	0.030	0.018	0.0932
No	Ref	-	-	Ref	-	-
BMI (kg/m^2^)	−0.022	0.001	<0.0001	0.042	0.001	<0.0001
Cholesterol (mg/dL)	0.002	0.000	<0.0001	0.005	0.000	<0.0001
Total energy intake (kcal)	0.010	0.002	<0.0001	0.001	0.005	0.8851
Year	−0.002	0.001	0.1116	−0.004	0.003	0.1404

**Table 5 ijerph-17-05769-t005:** Results of subgroup multiple regression analysis for the relationship between active use of nutrition labelling and outcome variables according to body-size perception *.

Body-Size Perception	Active Use of Nutrition Labelling	HDL-C (mg/dL)			TG (mg/dL)		
		β	SE	*p*-Value	β	SE	*p*-Value
Misperception (Over)	Check and make dependent purchase decisions	0.010	0.014	0.4438	−0.036	0.031	0.2457
Misperception (Under)	Check and make dependent purchase decisions	0.010	0.009	0.3019	−0.031	0.021	0.1430
Healthy Perception	Check and make dependent purchase decisions	0.013	0.005	0.0100	−0.033	0.013	0.0105

* Adjusted for sex, age, educational level, economic status, household income, marital status, residence area, subjective health status, stress awareness, alcohol intake, smoking status, aerobic exercise habits, walking for more than 10 min, frequency of eating out, family history for hyperlipidemia, survey year, BMI, total energy intake, and cholesterol level.

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
