# Peer review of "Nutrition Labeling Usage Influences Blood Markers in Body-Size Self-Conscious Individuals: The Korean National Health and Nutrition Examination Survey (KNHANES) 2013–2018"

_ijerph, 2020, doi:10.3390/ijerph17165769_

Round 1
Reviewer 1 Report
This study by Kye et alia aims at assessing how labelling on food items can influence consumer’s decision, together with HDL-C and TG measurements. Besides, they also evaluated this effect in perspective of body size perception. In this study, the authors provide insights that food labelling is, indeed, efficient, but only if subjects have an adequate representation of their own body size.
The manuscript is well written and easy to read. Healthcare professionals, especially in nutrition, would find this study very useful. Moreover, governmental policies are here under scrutiny. The authors also found that such policies (food item labelling) can be effective to improve HDL-C and TG levels in consumers, but not for everyone. While the authors bring interesting new evidence, some (minor) changes are required in the manuscript to improve understanding. These changes are detailed below.
Intro
Line 58-61, please limit the use of “and”.
There is only one sentence related to HDL-C and TG. Authors should explain in more details the rationale for focusing on HDL-C and TG, since this is the main output of their study. Moreover, authors should explicit the ranges for healthy HDL-C and TG levels and the metabolism of both. Besides, how do HDL-C and TG parallel health/pathology ?
Methods
Please provide a Figure outlining the selection process for this study (with n numbers for initial screening, loss to follow up, excluded participants, etc). For an example, please see Khan et al., Nutrients, 2020 Apr, 12(4): 1196 (PMID 32344617).
In section 2.1.1 on the measurements of HDL-C and TG, authors should specify how these were measured (blood/serum analysis?) and using which measurement method(s)/kit(s). The discussion section (4.1) does mention results of blood tests. Please include such a protocol in the method section and the details on how this was performed (who, where, when and in what conditions [fasting?], venous/arterial blood ?).
Results
Table 1 is well presented. However, to my opinion, it would be interesting to separate men and women, at least concerning alcohol intake, smoking and body size perception.
Table 1 should be truncated into two separate tables. The first table should only present N numbers and %, while the second one should present Mean and SD (BMI, Waist circumference, Carbohydrate intake, Energy intake and Cholesterol). And again, how was “cholesterol (mg/dL)” measured ? (this should be indicated in the methods section).
In the result section explaining Table 2, on lines 146-152, authors mention that “Female sex, younger age […]” are positively or negatively associated to HDL-C or TG. To my opinion, such a grammatical formulation is confusing. Authors should use more direct comparisons. An example would be “HDL-C levels were significantly lower/higher in men/women, in younger/older subjects, etc…”.
In Table 3, authors should give more details concerning their analyses. Indeed, this section (on the bottom of page 7) lacks explanations on the calculations. What does the β calculation refer to ? This should also be mentioned in the methods section.
Discussion
In the Discussion section, whole chunks of text are duplicated. This concerns sections that begin with:
a/ “This study aimed to analyze”,
b/ “Responders who checked nutrition labelling” and
c/ “Our subgroup analysis regarding”
In section 4.1 on strengths and limitations, authors mention that the current study “focused on whether nutrition labelling had a real effect on food purchase, thus providing more practical data”. However, nowhere do authors present such results:
A/ If food purchasing was quantified, then authors should present such interesting results.
B/ If such an analysis is not possible, then authors should be more careful regarding that conclusion (food purchase). Authors can indeed infer that participants who take into account labelling are likely choosing healthier food items (which, in consequence tend to increase HDL-C and decrease TG), since a direct quantification was not made in the present study.
Author Response
Thank you for the comments regarding our manuscript. It is our pleasure to submit a new version of our paper that takes the reviewers’ critiques into account. We carefully considered and addressed each point raised by the reviewers.
Dear Editor:
Thank you for the comments regarding our manuscript. It is our pleasure to submit a new version of our paper that takes the reviewers’ critiques into account. We carefully considered and addressed each point raised by the reviewers. Below, reviewer comment is followed by the authors’ response. Modifications made to the manuscript are indicated by yellow highlighting.
Thank you very much for your interest and careful review of our study.
Responses to the review by reviewer #1:
Comment 1) Line 58-61, please limit the use of “and”.
<Response 1>
Thank you for your comment. We corrected the sentence.
[Page 2, line 68-71]
Distorted body images cause psychological problems such as over-exercising, eating disorders, decline in self-esteem, depression, and physical problems such as malnutrition, osteoporosis, digestive problems, and cardiovascular diseases [18,19].
Comment 2) There is only one sentence related to HDL-C and TG. Authors should explain in more details the rationale for focusing on HDL-C and TG, since this is the main output of their study. Moreover, authors should explicit the ranges for healthy HDLC and TG levels and the metabolism of both. Besides, how do HDL-C and TG parallel health/pathology ?
<Response 2>
Thank you for your feedback. We added the rationale for focusing on HDL-C and TG in more detail.
[Page 2, line 77-82]
Dyslipidemia is a state of abnormal amounts of lipids in the blood and is characterized by conditions such as decreased HDL-C (£40 mg/dL) and hypertriglyceridemia (³150 mg/dL) [24]. Although not necessarily harmful itself, the condition is a major risk factor for various cardiovascular diseases. Plasma concentrations of HDL-C have shown a strong inverse relationship with the risk of developing coronary artery disease and cancer [25]. Conversely, a low concentration of HDL-C is associated with obesity [25].
Comment 3) Please provide a Figure outlining the selection process for this study (with n numbers for initial screening, loss to follow up, excluded participants, etc). For an example, please see Khan et al., Nutrients, 2020 Apr, 12(4): 1196 (PMID 32344617).
<Response 3>
Thank you for your comment. Regarding your comment, we added the figure about selection process of study population as follows. Also, added following sentences in method section.
[Page 3, line 104-108]
We excluded respondents diagnosed with hypertension, diabetes, or dyslipidemia from the study (n=8,065/ n=3,163/ n=5,621), and those with missing values for the diagnosis of the diseases (n=558). Additionally, individuals not tested for dyslipidemia indicators such as HDL-C and TG or those with no response for independent variables were excluded(n=5,922).
Comment 4) In section 2.1.1 on the measurements of HDL-C and TG, authors should specify how these were measured (blood/serum analysis?) and using which measurement method(s)/kit(s). The discussion section (4.1) does mention results of blood tests. Please include such a protocol in the
method section and the details on how this was performed (who, where, when and in what conditions [fasting?], venous/arterial blood ?).
<Response 4>
Thank you for your comment. We added the measurements of HDL-C and TG in detail.
[Page 3, line 121-126]
KNHANES collected data through household member confirmation surveys, health questionnaire surveys, examination surveys and nutrition surveys. The examination surveys consisted of physical measurement, blood pressure and pulse measurement, blood and urine test, oral examination, lung function test, eye test, and grip test. The tests were performed on the day of the survey. The blood test conducted was a fasting blood test (minimum 8 h and recommended 12 h after eating), performed by a trained nurse in the mobile examination vehicle.
Comment 5) Table 1 is well presented. However, to my opinion, it would be interesting to separate men and women, at least concerning alcohol intake, smoking and body size perception.
<Response 5>
Thank you for your feedback. We re-organized Table 1 by sex.
[Page 5]
|
Variables |
Study population |
Male |
Female |
|||
|
N |
% |
N |
% |
N |
% |
|
|
Active use of nutrition labeling |
||||||
|
Check and make dependent purchase decisions |
4,308 |
26.04 |
1,052 |
16.81 |
3,256 |
35.06 |
|
Not actively using |
12,311 |
73.96 |
6,075 |
83.19 |
6,236 |
64.94 |
|
Sex |
||||||
|
Male |
7,127 |
49.45 |
7,127 |
100.00 |
|
|
|
Female |
9,492 |
50.55 |
|
|
9,492 |
100.00 |
|
Age (years) |
||||||
|
<20 |
264 |
1.94 |
140 |
2.13 |
124 |
1.76 |
|
<30 |
2,735 |
22.67 |
1,211 |
24.13 |
1,524 |
21.24 |
|
<40 |
4,058 |
25.42 |
1,591 |
24.45 |
2,467 |
26.36 |
|
<50 |
4,024 |
24.02 |
1,505 |
21.85 |
2,519 |
26.15 |
|
<60 |
2,942 |
16.34 |
1,233 |
16.40 |
1,709 |
16.29 |
|
<70 |
1,669 |
6.52 |
874 |
7.28 |
795 |
5.77 |
|
70+ |
927 |
3.09 |
573 |
3.77 |
354 |
2.42 |
|
Educational level |
||||||
|
High school or below |
7,294 |
39.25 |
2,958 |
35.67 |
4,336 |
42.75 |
|
Bachelor’s degree |
8,325 |
54.58 |
3,668 |
57.21 |
4,657 |
52.01 |
|
Master’s degree and above |
1,000 |
6.17 |
501 |
7.12 |
499 |
5.24 |
|
Economic status |
||||||
|
Employed |
5,556 |
31.87 |
1,587 |
21.38 |
3,969 |
42.12 |
|
Unemployed |
11,063 |
68.13 |
5,540 |
78.62 |
5,523 |
57.88 |
|
Household income |
||||||
|
Low |
3,844 |
24.05 |
1,638 |
23.76 |
2,206 |
24.33 |
|
Mid-low |
4,200 |
25.43 |
1,819 |
25.95 |
2,381 |
24.91 |
|
Mid-high |
4,282 |
25.48 |
1,848 |
25.42 |
2,434 |
25.54 |
|
High |
4,293 |
25.04 |
1,822 |
24.87 |
2,471 |
25.21 |
|
Marital status |
||||||
|
Single |
3,776 |
29.69 |
1,941 |
35.32 |
1,835 |
24.19 |
|
Separated/divorced/bereavement |
1,261 |
5.93 |
346 |
3.86 |
915 |
7.95 |
|
Married |
11,582 |
64.38 |
4,840 |
60.83 |
6,742 |
67.86 |
|
Residence Area |
||||||
|
Metropolitan |
9,960 |
60.29 |
4,178 |
59.11 |
5,782 |
61.45 |
|
Others |
6,659 |
39.71 |
2,949 |
40.89 |
3,710 |
38.55 |
|
Perceived health status |
||||||
|
Good |
14,484 |
87.43 |
6,341 |
89.09 |
8,143 |
85.80 |
|
Bad |
2,135 |
12.58 |
786 |
10.91 |
1,349 |
14.20 |
|
Stress awareness |
||||||
|
High |
4,524 |
28.36 |
1,762 |
26.24 |
2,762 |
30.43 |
|
Low |
12,095 |
71.64 |
5,365 |
73.76 |
6,730 |
69.57 |
|
Alcohol intake |
||||||
|
Less than 1 time per month |
6,176 |
34.18 |
1,739 |
23.41 |
4,437 |
44.72 |
|
1–3 times per Week |
9,402 |
59.6 |
4,605 |
66.83 |
4,797 |
52.53 |
|
More than 4 times per Week |
1,041 |
6.21 |
783 |
9.76 |
258 |
2.75 |
|
Smoking status |
||||||
|
Smoker |
3,305 |
23.16 |
2,743 |
40.03 |
562 |
6.64 |
|
Ex-smoker |
3,256 |
19.77 |
2,607 |
32.76 |
649 |
7.06 |
|
Non-smoker |
10,058 |
57.08 |
1,777 |
27.21 |
8,281 |
86.30 |
|
Aerobic exercise habits |
||||||
|
Yes |
7,539 |
48.34 |
3,432 |
48.56 |
4,107 |
45.31 |
|
No |
9,080 |
51.66 |
3,695 |
51.44 |
5,385 |
54.69 |
|
Walking for more than 10 minutes |
||||||
|
Few |
2,515 |
14.27 |
1,157 |
15.24 |
1,358 |
13.32 |
|
1–4 days per week |
5,133 |
30.18 |
2,109 |
28.97 |
3,024 |
31.36 |
|
5–7 days per week |
8,971 |
55.55 |
3,861 |
55.80 |
5,110 |
55.31 |
|
Frequency of eat out |
||||||
|
More than 4 times per week |
9,762 |
63.83 |
5,111 |
76.38 |
4,841 |
48.45 |
|
Less than 3 times per week |
6,857 |
36.17 |
2,016 |
23.62 |
4,651 |
51.55 |
|
Family history of hyperlipidemia |
||||||
|
Yes |
1,158 |
7.01 |
368 |
5.63 |
790 |
8.36 |
|
No |
15,461 |
92.99 |
6,759 |
94.37 |
8,702 |
91.64 |
|
Year |
||||||
|
2013 |
2,715 |
15.94 |
1,173 |
16.18 |
1,542 |
15.69 |
|
2014 |
2,491 |
15.57 |
1,034 |
15.53 |
1,457 |
15.60 |
|
2015 |
2,564 |
16.52 |
1,126 |
16.54 |
1,438 |
16.49 |
|
2016 |
2,854 |
16.55 |
1,177 |
16.17 |
1,677 |
16.94 |
|
2017 |
2,917 |
17.17 |
1,263 |
16.94 |
1,654 |
17.39 |
|
2018 |
3,078 |
18.25 |
1,354 |
18.64 |
1,724 |
17.88 |
|
Body size perception |
||||||
|
Misperception (over) |
3,051 |
18.53 |
1,822 |
24.30 |
824 |
8.07 |
|
Misperception (under) |
2,646 |
15.11 |
613 |
9.11 |
2,438 |
26.53 |
|
Healthy perception |
10,922 |
66.36 |
4,692 |
66.60 |
6,230 |
65.40 |
|
Total |
16,619 |
100.00 |
7,127 |
100.00 |
9,492 |
100.00 |
Comment 6) Table 1 should be truncated into two separate tables. The first table should only present N numbers and %, while the second one should present Mean and SD (BMI, Waist circumference, Carbohydrate intake, Energy intake and Cholesterol). And again, how was “cholesterol (mg/dL)” measured ? (this should be indicated in the methods section).
<Response 6>
Thank you for your feedback. Based on your comment, we decided to separate Table 1 as two types of table by the characteristics of variable. We also added related paragraph in result section as follows. “Average of BMI and waist circumference in this study were 23.45 and 80.12. Its values are higher in male than female. The respondents have intake total energy about 2150 Kcal in day, and the proportion of carbohydrate among total energy intake was 59.81%. Female had intake low calories in daily, but had higher proportion of carbohydrate among total energy. The total cholesterol was average 191.20, and it was similar by sex (Table 2).”
KNHANES collects survey data through household member confirmation surveys, health questionnaire surveys, examination surveys and nutrition surveys. The examination surveys consisted of physical measurement, blood pressure and pulse measurement, blood and urine test, oral examination, lung function test, eye test, and grip test. The test was performed at day of survey. As your comment, we also demonstrated the method about measuring blood markers.
[Page 3, line 121-126]
KNHANES collected data through household member confirmation surveys, health questionnaire surveys, examination surveys and nutrition surveys. The examination surveys consisted of physical measurement, blood pressure and pulse measurement, blood and urine test, oral examination, lung function test, eye test, and grip test. The tests were performed on the day of the survey. The blood test conducted was a fasting blood test (minimum 8 h and recommended 12 h after eating), performed by a trained nurse in the mobile examination vehicle
[Page 8, line 185-190]
The average BMI and waist circumference in this study were 23.45 kg/m2 and 80.12 cm, respectively. The values were higher in males than females. The respondents had a total energy intake of about 2,150 Kcal in a day, and the proportion of carbohydrates in the total energy intake was 59.81%. Females had a lower intake of calories in a day, but had a higher proportion of carbohydrates in the total energy intake. The average total cholesterol was 191.20 mg, and it was similar in both sexes (Table 2).
Table 2. General characteristics of the study population.
|
Variables |
Study population |
Male |
Female |
|||
|
Mean |
SD |
Mean |
SD |
Mean |
SD |
|
|
BMI (kg/m2) |
23.45 |
0.03 |
24.28 |
0.05 |
22.64 |
0.05 |
|
Waist circumference (cm) |
80.12 |
0.10 |
84.58 |
0.13 |
75.75 |
0.13 |
|
Proportion of carbohydrate among total energy intake (%) |
59.81 |
0.14 |
57.88 |
0.20 |
61.70 |
0.16 |
|
Total energy intake (Kcal) |
2,145.85 |
9.88 |
2,508.22 |
15.48 |
1,791.34 |
8.76 |
|
Cholesterol (mg) |
191.20 |
0.33 |
192.14 |
0.47 |
190.29 |
0.41 |
Comment 7) In the result section explaining Table 2, on lines 146-152, authors mention that “Female sex, younger age […]” are positively or negatively associated to HDL-C or TG. To my opinion, such a grammatical formulation is confusing. Authors should use more direct comparisons. An example would be “HDL-C levels were significantly lower/higher in men/women, in younger/older subjects, etc…”.
<Response 7>
Thank you for your feedback. We have modified the sentences as you advised.
[Page 8, line 197-205]
HDL-C levels were significantly higher in respondents with female sex (p < .0001), younger age (p = 0.0008), higher household income (p = 0.0319), single status (p < .0001), better perceived health status (p = 0.0004), low stress awareness (p = 0.0138), higher alcohol intake (p < .0001), non-smoking status (p < .0001), aerobic exercise habits (p = 0.0007), or frequent walking for more than 10 minutes (p < .0001). TG levels were significantly lower in participants with female sex (p < .0001), younger age (p < .0001), bachelor’s degree (p = 0.0002), single status (p = 0.0087), low stress awareness (p = 0.0133), lower alcohol intake (p < .0001), non-smoking status (p < .0001), aerobic exercise habits (p = 0.0003), frequent walking for more than 10 minutes (p = 0.0125), or no family history of hyperlipidemia (p = 0.0141).
Comment 8) In Table 3, authors should give more details concerning their analyses. Indeed, this section (on the bottom of page 7) lacks explanations on the calculations. What does the β calculation
refer to ? This should also be mentioned in the methods section.
<Response 8>
Thank you for your feedback. In this study, we performed the linear regression analysis to investigate the relationship between HDL-C or TG and nutrition labelling by perception their body size. We considered the outcome variables which were log-transformed for controlling their distribution. Thus, the results could be explained as ## percent increased by independent variable, after exponentiating the coefficient values and multiplied by 100. Regarding your comment, we revised method and result section as follows.
[Page 4, line 168-170]
Considering the distribution of outcome variables, we log-transformed them; the results of regression analysis could be interpreted as percent changes after exponentiating the coefficients and multiplying by 100.
[Page 11, line 213-215]
Table 4 shows results of multiple linear regression to examine the association between active use of nutrition labeling and outcome variables. Individuals who used nutrition labeling actively had significantly higher HDL-C (0.95% increases) and lower TG levels (2.83% decreases) (β = 0.009, p = 0.0317; β = -0.029, p = 0.0073).
[Page 13, line 218-221]
We also conducted subgroup multiple linear regression analysis to investigate relationships between active use of nutrition labeling and outcome variables according to body size perception. In the group that comprised participants who had correct awareness of their body size, the participants who actively used nutrition labeling had higher HDL-C (1.12% increases) and lower TG levels (-2.93% decreases) than those who did not actively use nutrition labeling (β = 0.011, p = 0.0315; β = -0.030, p = 0.0203).
Comment 9) In the Discussion section, whole chunks of text are duplicated. This concerns sections that begin with: a/ “This study aimed to analyze”, b/ “Responders who checked nutrition labelling” and
c/ “Our subgroup analysis regarding”
<Response 9>
Thank you for your feedback. We deleted repeated paragraphs.
Comment 10) In section 4.1 on strengths and limitations, authors mention that the current study “focused on whether nutrition labelling had a real effect on food purchase, thus providing more
practical data”. However, nowhere do authors present such results:
A/ If food purchasing was quantified, then authors should present such interesting results.
B/ If such an analysis is not possible, then authors should be more careful regarding that conclusion (food purchase).
Authors can indeed infer that participants who take into account labelling are likely choosing healthier food items (which, in consequence tend to increase HDL-C and decrease TG), since a direct quantification was not made in the present study.
<Response 10>
Thank you for your comment. We totally agreed with your advice. We deleted the sentences.

Reviewer 2 Report
Dear authors, I find the topic of your manuscript very interesting and the potential repercussions on public health issues are very relevant. However, some aspects deserve further insight and clarification and that could improve the reporting of your work.
- As for the title, and later also the abstract, I found it difficult to understand that the correlation you spoke about regarded two concomitant conditions: active evaluation of food labels in conjunction with the correct perception of body weight. I advise you to better rephrase this aspect which is currently clarified only after reading the full text. For example, the title could be reworded in: "Nutrition labels usage influences blood markers in body size self-conscious individuals: the Korean ..." or something similar. In the abstract, "the use of nutrition labelling by body size perception" is not very clear.
- Why LDL-C, that represent an important marker for metabolic pathologies, have not been evaluated?
- It might be useful to add some references to line 47, especially since these are limited data present in the literature.
- Similarly, some examples (ref) could be added to line 64 to clarify the aspects referred to
- Lines 49-57 seem slightly offtopic since the theme of personalization is not consistent with the hypotheses made in the manuscript which, on the other hand, try to clarify aspects that can be generalized for the population.
- On line 70, are you sure that the phrase "differences in the intake levels" is correct?
- Line 73 could be reworded to be clearer, as well as proposed for title and abstract
- Line 81 could indicate the starting number of subject in the sample before exclusions
- On line 103 it is better to change the term "subjective" to "perceived"
- As regards the correct perception of body weight, have any tolerance criteria been used? How many BMI points of tolerance? It should be indicated in the materials
- Furthermore, how were the perceived levels of stress and health measured? Did you use any validated criteria?
- Line 116 refers to an obesity cutoff above the value of 25. In other countries, this is the value of the overweight, a condition not mentioned in the manuscript. Can you please add a bibliographic reference about the cutoffs used in South Korea?
- In the manuscript, it would be better to refer to HDL-C and TG as blood markers rather than outcomes.
- There is a little confusion between awareness and active use of nutrition labelling. Please standardize the text through the manuscript
- Table 2 lacks association values. Which statistical entity was used? How can the reader evaluate the direction of the association (direct or inverse) without a value?
- On page 7 it is specified that the association with markers is advantageous in case of healthy behaviour. How do you explain the association with alcohol (which also represents the strongest association of those found in table 3)?
- At the last two lines of page 9, did you mean "not significant"?
- Table 4 is the core of the manuscript, although the association remains modest. Have you tested the effect on blood markers in case of correct perception of body weight but lack of attention to nutritional labels? Is there a correlation between correct body perception and HDL-C / TG? Is there a correlation between correct perception of body weight and attention to nutrition labels? These aspects would help to better understand the phenomenon proposed in the manuscript.
- On page 11, considering the results, there is a slight overinterpretation of the phenomenon. I recommend a more cautious approach.
- Among the limitations, I would add all those aspects that are self-reported (stress, health status, use of labels for food choices, etc.).
Formatting aspects and graphic typos
- Numbering problems from page 11 onwards. The parts of text written after the tables should be separated from them and inserted into a vertical sheet. Moreover, from page 7, the lines are no longer numbered and it is not possible to accurately indicate a specific passage for revisions.
- Extra space on line 151.
- In the discussion, a long paragraph (starting from line 1 of page 11 until almost the end of the page), is repeated two-times. Please, cut the repetition
Author Response
Thank you for the comments regarding our manuscript. It is our pleasure to submit a new version of our paper that takes the reviewers’ critiques into account. We carefully considered and addressed each point raised by the reviewers.
Thank you for the comments regarding our manuscript. It is our pleasure to submit a new version of our paper that takes the reviewers’ critiques into account. We carefully considered and addressed each point raised by the reviewers. Below, reviewer comment is followed by the authors’ response. Modifications made to the manuscript are indicated by yellow highlighting.
Thank you very much for your interest and careful review of our study.
Responses to the review by reviewer #2:
Comment 1) As for the title, and later also the abstract, I found it difficult to understand that the correlation you spoke about regarded two concomitant conditions: active evaluation of food labels in
conjunction with the correct perception of body weight. I advise you to better rephrase this aspect which is currently clarified only after reading the full text. For example, the title could be reworded in: "Nutrition labels usage influences blood markers in body size self-conscious individuals: the Korean
..." or something similar. In the abstract, "the use of nutrition labelling by body size perception" is not very clear.
<Response 1>
Thank you for your advice. We reworded the title as you advised and corrected the abstract.
[Page 1, line 2-7]
Nutrition labeling usage influences blood markers in body size self-conscious individuals: The Korean National Health and Nutrition Examination Survey (KNHANES) 2013–2018
[Page 1, line 13-15]
This study analyzed the effects of nutrition labeling and examined whether nutrition labeling usage influences the levels of blood markers such as high-density lipoprotein cholesterol (HDL-C) and triglyceride (TG) in body size self-conscious individuals.
Comment 2) Why LDL-C, that represent an important marker for metabolic pathologies, have not been evaluated?
<Response 2>
Thank you for your question. We are totally agreed with your comment, but in KNHANES, the LDL-C was not collected during same period like as HDL-C or TG. In KNHANES during 2013-2018, LDL-C was only collected from patients who had 200mg/dL of TG. Thus, the results for LDL-C could not have the external validity, not like other indicators about metabolism, and we did not include that variable. Of course, based on Friedwald Formula, we could calculate this value as alternatively, but Its results had not statistically significant like as previous studies that healthy behavior major associated with TG or HDL-C. If needed, we willing add those results as follows.
|
Variables |
LDL cholesterol (mg/dL) |
||
|
β |
SE |
P-value |
|
|
Awareness on nutrition labelling |
|||
|
Check and make dependent purchase decisions |
0.003 |
0.003 |
0.3624 |
|
Not actively use |
Ref |
- |
- |
Comment 3) It might be useful to add some references to line 47, especially since these are limited data present in the literature.
<Response 3>
Thank you for your feedback. We added the references.
[Page 2, line 56]
However, it is challenging to find studies that have categorized individuals according to varied personal characteristics in the existing literature, which mainly comprises studies conducted on general characteristics or involving individuals with specific diseases [9-13].
Comment 4) Similarly, some examples (ref) could be added to line 64 to clarify the aspects referred to
<Response 4>
Thank you for your comment. We added the references.
[Page 2, line 76]
Previous studies have focused on how intervention strategies affect body image but little attention has been paid towards research using body image as a resource to develop intervention strategies for healthy eating habits [22,23].
Comment 5) Lines 49-57 seem slightly offtopic since the theme of personalization is not consistent with the hypotheses made in the manuscript which, on the other hand, try to clarify aspects that can be generalized for the population
<Response 5>
Thank you for your comment. We have corrected the manuscript accordingly.
[Page 2, line 57-67]
To practice a healthy lifestyle, it is important to cultivate habits that increase health-related knowledge and beliefs, improve attitude, and enhance self-efficacy [14]. One of the intervention strategies to foster such a lifestyle is “tailoring.” Tailoring can be defined as any of the several methods applied for creating communications that are customized, with the expectation that there will be larger intended effects because of these communications [15]. Tailoring is generally considered to be a useful method of increasing the effectiveness of health interventions, through the delivery of tailored print, as well as telemedicine or eHealth applications [16]. Thus, it could be important to identify various personal factors for tailoring.
Comment 6) On line 70, are you sure that the phrase "differences in the intake levels" is correct?
<Response 6>
Thank you for your question. We corrected the sentence.
[Page 2, line 85-86]
whether there were differences in the levels of HDL-C and TG
Comment 7) Line 73 could be reworded to be clearer, as well as proposed for title and abstract
<Response 7>
Thank you for your feedback. We reworded the sentence.
[Page 2, line 89-90]
whether nutrition labeling usage influences the levels of HDL-C and TG in body size self-conscious individuals.
Comment 8) Line 81 could indicate the starting number of subject in the sample before exclusions
<Response 8>
Thank you for your comment. We added that in Figure 1 about selection criteria of study population, and revised method section.
Comment 9) On line 103 it is better to change the term "subjective" to "perceived"
<Response 9>
Thank you for your feedback. We changed the term as you advised.
[Page 4, line 135]
perceived health status
Comment 10) As regards the correct perception of body weight, have any tolerance criteria been used? How many BMI points of tolerance? It should be indicated in the materials
<Response 10>
Thank you for your feedback. When subjective body shape recognition was exaggerated in comparison with the actual body shape, body size perception was defined as misperception (over), whereas it was defined as misperception (under) when subjective body shape recognition was understated. When participants perceived their actual body shape correctly, body size perception was defined as correct perception.
Comment 11) Furthermore, how were the perceived levels of stress and health measured? Did you use any validated criteria?
<Response 11>
Thank you for your question. We measured the variables based on self-reported survey without any validated criteria.
[Page 4, line 141-142]
Perceived health status and stress awareness were categorized as good/high or bad/low.
Comment 12) Line 116 refers to an obesity cutoff above the value of 25. In other countries, this is the value of the overweight, a condition not mentioned in the manuscript Can you please add a not mentioned in the manuscript. Can you please add a bibliographic reference about the cutoffs used in South Korea?
<Response 12>
Thank you for your comment. We added the explanation and a reference.
[Page 4, line 148-152]
BMI was defined as underweight, normal, and obese based on obesity criteria in South Korea (<18.5, 18.5–25, and >25, respectively) [32]. Adult obesity is generally defined as BMI ≥ 30kg/m2 in Western populations, while overweight is defined as BMI between 25 and 29.9 kg/m2 [28]. However, lower BMI cutoff points apply to other populations, including East Asians.
Comment 13) In the manuscript, it would be better to refer to HDL-C and TG as blood markers rather than outcomes
<Response 13>
Thank you for your feedback. We added the rationale for focusing on HDL-C and TG in more detail.
[Page 2, line 77-82]
Dyslipidemia is a state of abnormal amounts of lipids in the blood and is characterized by conditions such as decreased HDL-C (£40 mg/dL) and hypertriglyceridemia (³150 mg/dL) [24]. Although not necessarily harmful itself, the condition is a major risk factor for various cardiovascular diseases. Plasma concentrations of HDL-C have shown a strong inverse relationship with the risk of developing coronary artery disease and cancer [25]. Conversely, a low concentration of HDL-C is associated with obesity [25].
Comment 14) There is a little confusion between awareness and active use of nutrition labelling. Please standardize the text through the manuscript
<Response 14>
Thank you for your feedback. We standardized the text through the manuscript with term “active use”
Comment 15) Table 2 lacks association values. Which statistical entity was used? How can the reader evaluate the direction of the association (direct or inverse) without a value?
<Response 15>
Thank you for your feedback. In table 2, the results about Analysis of variance to compare the average and standard deviation of outcome variables by the independent variables were showed. To assist reader`s interpretation, we revised result section.
[Page 8, line 193-195]
Table 3 shows results of analysis of variance to examine associations between the independent and outcome variables to compare the association of HDL-C and TG levels with the categorical variables.
Comment 16) On page 7 it is specified that the association with markers is advantageous in case of healthy behaviour. How do you explain the association with alcohol (which also represents the strongest association of those found in table 3)?
<Response 16>
Thank you for your question. We corrected the sentence as follows. Previous studies also showed that HDL-C levels were significantly higher in respondents with higher alcohol intake [kim et al, 2015; Yoo et al., 2019]. We assume that the moderate amount of alcohol could cause a higher level of HDL-C.
- Kim, J.Y.; Kweon, K.H.; Kim, M.J.; Park, E.-C.; Jang, S.-Y.; Kim, W.; Han, K.-T. Is nutritional labeling associated with individual health? The effects of labeling-based awareness on dyslipidemia risk in a South Korean population. Nutr J 2015, 15. https://doi.org/10.1186/s12937-016-0200-y
- Yoo, J.S.; Han, K.T.; Chung, S.H.; Park, E.C. Association between awareness of nutrition labeling and high-density lipoprotein cholesterol concentration in cancer survivors and the general population: The Korean National Health and Nutrition Examination Survey (KNHANES) 2010-2016. BMC Cancer 2019, 19, 16. https://doi.org/10.1186/s12885-018-5196-6
[Page 11, Line 216]
Male or older individuals generally had higher risk levels associated with the two indicators, whereas individuals with healthy behaviors had lower risk levels associated with the two indicators, with the exception of alcohol intake.
Comment 17) At the last two lines of page 9, did you mean "not significant"?
<Response 17>
Thank you for your feedback. We corrected the sentence.
[Page 13, Line 222]
In contrast, in the group that comprised participants who did not have correct awareness of their body size, there were not significant differences in HDL-C and TG levels according to the active use of nutrition labeling
Comment 18) Table 4 is the core of the manuscript, although the association remains modest. Have you tested the effect on blood markers in case of correct perception of body weight but lack of attention to nutritional labels? Is there a correlation between correct body perception and HDL-C / TG? Is there a correlation between correct perception of body weight and attention to nutrition labels? These aspects would help to better understand the phenomenon proposed in the manuscript.
<Response 18>
Thank you for your feedback. In sub-group analyses, we tested the association between attention for nutrition labelling and blood markers by the perception of body size. At the results of sub-group as correct perception, respondents with higher awareness on nutrition labelling had statistically healthy results than low awareness people who you have questioned.
Comment 19) On page 11, considering the results, there is a slight overinterpretation of the phenomenon. I recommend a more cautious approach.
<Response 19>
Thank you for your feedback. We have corrected the manuscript accordingly.
[Page 15, line 245-249]
As a result of active use of nutrition labeling, healthy dietary decisions were made by the participants while purchasing food and therefore it appears that nutrition-related biomarkers such as HDL-C and TG could be influenced by nutrition labeling. In other words, the introduction of a nutrition labeling system could be beneficial in increasing health behavior, reducing obesity, and improving health outcomes [33].
[Page 15, line 250-262]
Our subgroup analysis regarding body size perception revealed interesting findings. The results of regression analysis after adjusting for almost all variables, revealed that the group of participants that recognized their body image correctly showed high HDL-C and low TG levels when they actively used nutrition labeling, but the other group of participants that did not correctly recognize their body size and overestimated or underestimated their body shape did not show significant changes in HDL-C and TG levels even if they actively use nutrition labeling. This shows that nutrition information does not play an appropriate role if people do not recognize their body size correctly, indicating that exact recognition of individual body image is important for healthy eating. A significant predictor of disturbed eating attitudes and behaviors was body image dissatisfaction, indicating that a negative body image perception was one of the key factors contributing to eating disorders [34,35]. In addition, standard nutritional information that does not fit one’s body size perception could be inefficient, and when not tailored to individuals, this information could not be expected to yield maximum benefits.
Comment 20) Among the limitations, I would add all those aspects that are self-reported (stress, health status, use of labels for food choices, etc.).
<Response 20>
Thank you for your comment. We added the limitation.
[Page 15, line 326-328]
Second, active use of nutrition labeling, perceived health status, and stress awareness were assessed using a self-reported survey, which raises issues related to recall bias of the participants.
Reviewer 3 Report
Thank you for a well prepared manuscript. Please find below some general feedback and recommended line by line revisions.
General comments
Your results read like correlations, when in fact you've performed ANOVA and multiple linear regressions. This needs to be amended with a more conventional reporting style in text, to support the extensive tables.
Please delete repeated paragraphs!
I would like you to expand your discussion, you have used a very rich dataset and at present the conclusion does not reflect the richness in the data you have analysed nor the results you have reported. This may be to accounting for most variables in the multiple linear regression
Line by line revisions
36/37 - I'm unsure that this sentence is an appropriate way to end this paragraph, it may sit better at the start of the next paragraph.
41/42 - Is incorrect labelling unethical or misleading?
49/57 - I understand the scope of this paragraph, and the intention behind it, but the latter half needs some rewording and clarification. I agree with your definition of tailoring and its potential impacts, but tailoring is prevalent in the real world dependent upon sector e.g. online sales such as Amazon, or food delivery services, and even as your references allude to in health services. Please amend this. Please also refine the sentence that includes reference 16
61 - 'wastage of resources' feel insensitive here since you've just outlined some rather serious health consequences. It is also not clear what is meant by this term when used in this manner. Please amend for clarification.
61/62 - would healthy not perhaps be a more appropriate way to describe a perception of body image as opposed to correct?
Please expand upon the nature of your multiple linear regression, was it forced or forward, backward, stepwise? This is important as it effects how results are to be reported.
144 - please check text formatting as this currently looks like an addition/note to Table 1. Likewise for discussion pertaining to Table 3.
146 - please provide outcome variables to accompany p values. Please also report ANOVA effects in full. It may also be wise to describe the relationship as non-significant as opposed to stating no relationship, as you have a p value of ~ 0.06 which approaches statistical significance.
Table 4 - I'm not sure what this table adds. This may work better if the supporting text is simply expanded for clarification. I may be wrong here though, as the table is well presented.
Discussion - please note page numbering has changed
'This finding is in line with those of previous studies' - please reference said studies
'...significant changes in HDL-C and TG' - you didn't measure change scores but used a multiple linear regression that was adjusted for almost all variables to assess the relationship between perception of body shape and HDL-C and TG. The way this is currently written suggests a causative as opposed to a predictive relationship, please amend.
'although most health information is overwhelming' - this is a strong generalisation. It may be that most health information is perceived as overwhelming because it is either poorly communicated or is not tailored to be meaningful at an individual level - the latter further supports your introductory comments
Please delete repeated paragraphs!
Author Response
Thank you for the comments regarding our manuscript. It is our pleasure to submit a new version of our paper that takes the reviewers’ critiques into account. We carefully considered and addressed each point raised by the reviewers.
Dear Editor:
Thank you for the comments regarding our manuscript. It is our pleasure to submit a new version of our paper that takes the reviewers’ critiques into account. We carefully considered and addressed each point raised by the reviewers. Below, reviewer comment is followed by the authors’ response. Modifications made to the manuscript are indicated by yellow highlighting.
Thank you very much for your interest and careful review of our study.
Responses to the review by reviewer #3:
Comment 1) 36/37 - I'm unsure that this sentence is an appropriate way to end this paragraph, it may sit better at the start of the next paragraph.
<Response 1>
Thank you for your comment. We moved the sentence as you advised.
Comment 2) 41/42 - Is incorrect labelling unethical or misleading?
<Response 2>
Thank you for your comment. We changed the term.
[Page 2, line 48-49]
Labeling also prevents consumers from succumbing to misleading advertisements on the nutritious value of food products, as accurate nutritional information is provided.
Comment 3) 49/57 - I understand the scope of this paragraph, and the intention behind it, but the latter half needs some rewording and clarification. I agree with your definition of tailoring and its
potential impacts, but tailoring is prevalent in the real world dependent upon sector e.g. online sales such as Amazon, or food delivery services, and even as your references allude to in health services. Please amend this. Please also refine the sentence that includes reference 16
<Response 3>
Thank you for your feedback. We have corrected the manuscript accordingly.
[Page 2, line 57-67]
To practice a healthy lifestyle, it is important to cultivate habits that increase health-related knowledge and beliefs, improve attitude, and enhance self-efficacy [14]. One of the intervention strategies to foster such a lifestyle is “tailoring.” Tailoring can be defined as any of the several methods applied for creating communications that are customized, with the expectation that there will be larger intended effects because of these communications [15]. Tailoring is generally considered to be a useful method of increasing the effectiveness of health interventions, through the delivery of tailored print, as well as telemedicine or eHealth applications [16]. Thus, it could be important to identify various personal factors for tailoring.
Comment 4) 61 - 'wastage of resources' feel insensitive here since you've just outlined some rather serious health consequences. It is also not clear what is meant by this term when used in this manner. Please amend for clarification
<Response 4>
Thank you for your comment. We agreed with your advice. We deleted the part.
[Page 2, line 68-71]
Distorted body images cause psychological problems such as over-exercising, eating disorders, decline in self-esteem, depression, and physical problems such as malnutrition, osteoporosis, digestive problems, and cardiovascular diseases [18,19].
Comment 5) 61/62 - would healthy not perhaps be a more appropriate way to describe a perception of body image as opposed to correct?
<Response 5>
Thank you for your comment. We standardized the text through the manuscript with term “healthy”
Comment 6) Please expand upon the nature of your multiple linear regression, was it forced or forward, backward, stepwise? This is important as it effects how results are to be reported.
<Response 6>
Thank you for your comments. Actually, we originally selected the independent variable based on the relationship with blood marker in previous studies. However, based on your comment, we tested stepwise selection, and variables were selected as follows.
1) HDL-C
= Aerobic exercise habits, Age (years), Alcohol intake, Awareness on nutrition labelling, BMI(kg/m2), Cholesterol (mg), Marital status, Proportion of carbohydrate among total energy intake (%), Sex, Smoking status, Stress awareness, Subjective health status, Total energy intake (Kcal), Waist circumference (cm), Walking for more than 10 minutes, Year
2) TG
= Aerobic exercise habits, Age (years), Alcohol intake, Awareness on nutrition labelling, BMI(kg/m2), Cholesterol (mg), Educational level, Household income, Sex, Smoking status, Stress awareness, Waist circumference (cm), Walking for more than 10 minutes, Year
Economic status, Residence Area, Frequency of eat out, Family history for hyperlipidemia were not included based on statistical variable selection method. However, these variables are also substantially associated with dyslipidemia based on many previous studies. In addition, the direction and magnitude of regression analysis are not different between our model and statistical variable selection model. Thus, we added evidences which supported the four variables although these variables are not included by statistically selection criteria.
- Santo, L.R.E.; Faria, T.O.; Silva, C.S.O.; Xavier, L.A.; Reis, V.C.; Mota, G.A.; et al. Socioeconomic status and education level are associated with dyslipidemia in adults not taking lipid-lowering medication: a population-based study. Int health 2019, https://doi.org/10.1093/inthealth/ihz089.
- Shin, J.; Cho, K.H.; Choi, Y.; Lee, S.G.; Park, E.C.; Jang S.I. Combined effect of individual and neighborhood socioeconomic status on mortality in patients with newly diagnosed dyslipidemia: a nationwide Korean cohort study from 2002 to 2013. Nutr Metab Cardiovasc Dis 2016, 26, 207-215. https://doi.org/10.1016/j.numecd.2015.12.007.
- Choi, M.K.; Lee, Y.K.; Bae, Y.J. Association of eating out frequency with the risks of obesity, diabetes, and dyslipidemia among Korean adults. Current developments in nutrition 2019, 3. https://www.researchgate.net/deref/http%3A%2F%2Fdx.doi.org%2F10.1093%2Fcdn%2Fnzz039.P18-058-19.
- Jung, C.H.; Park, J.S.; Lee, W.Y.; Kim, S.W. Effects of smoking, alcohol, exercise, level of education, and family history on the metabolic syndrome in Korean adults. Korean J Med 2002, 63, 649-659. https://www.koreamed.org/SearchBasic.php?RID=1007KJM/2002.63.6.649&DT=1.
If needed, we will change the results by statistical selection criteria as follows.
|
Variables |
HDL cholesterol (mg/dL) |
Triglyceride (mg/dL) |
||||
|
β |
SE |
P-value |
β |
SE |
P-value |
|
|
Awareness on nutrition labelling |
||||||
|
Check and make dependent purchase decisions |
0.009 |
0.004 |
0.0330 |
-0.027 |
0.011 |
0.0115 |
|
Not actively use |
Ref |
- |
- |
Ref |
- |
- |
- The results of regression analysis adjusting Aerobic exercise habits, Age (years), Alcohol intake, BMI(kg/m2), Cholesterol (mg), Educational level, Household income, Marital status, Proportion of carbohydrate among total energy intake (%), Sex, Smoking status, Stress awareness, Subjective health status, Total energy intake (Kcal), Waist circumference (cm), Walking for more than 10 minutes, Year
Comment 7) 144 - please check text formatting as this currently looks like an addition/note to Table 1. Likewise for discussion pertaining to Table 3.
<Response 7>
Thank you for your comments. We corrected the sentence
[Page 8, line 193-195]
Table 3 shows results of analysis of variance to examine associations between the independent and outcome variables to compare the association of HDL-C and TG levels with the categorical variables.
Comment 8) 146 - please provide outcome variables to accompany p values. Please also report ANOVA effects in full. It may also be wise to describe the relationship as non-significant as
opposed to stating no relationship, as you have a p value of ~ 0.06 which approaches statistical significance.
<Response 8>
Thank you for your comments. We revised the sentences as you advised.
[Page 8, line 195–205]
There was non-significant relationship between active use of nutrition labeling and outcome variables (p = 0.2282; p = 0.0653). HDL-C levels were significantly higher in respondents with female sex (p < .0001), younger age (p = 0.0008), higher household income (p = 0.0319), single status (p < .0001), better perceived health status (p = 0.0004), low stress awareness (p = 0.0138), higher alcohol intake (p < .0001), non-smoking status (p < .0001), aerobic exercise habits (p = 0.0007), or frequent walking for more than 10 minutes (p < .0001). TG levels were significantly lower in participants with female sex (p < .0001), younger age (p < .0001), bachelor’s degree (p = 0.0002), single status (p = 0.0087), low stress awareness (p = 0.0133), lower alcohol intake (p < .0001), non-smoking status (p < .0001), aerobic exercise habits (p = 0.0003), frequent walking for more than 10 minutes (p = 0.0125), or no family history of hyperlipidemia (p = 0.0141).
Comment 9) Table 4 - I'm not sure what this table adds. This may work better if the supporting text is simply expanded for clarification. I may be wrong here though, as the table is well presented.
<Response 9>
Thank you for your comments. Table 4 shows results of subgroup multiple linear regression analysis to investigate whether nutrition labeling usage influences the levels of HDL-C and TG in body size self-conscious individuals. The group that recognized body image correctly showed high HDL-C and low TG levels when they actively used nutrition labeling, whereas the group that recognized body image incorrectly showed no significant changes in HDL-C and TG levels even when actively using nutrition labeling.
Comment 10) 'This finding is in line with those of previous studies' – please reference said studies
<Response 10>
Thank you for your comments. We described the references.
[Page 15, line 241]
This finding is in line with those of previous studies, indicating that the habits of reading nutrition information and applying it in real life could yield healthy outcomes, emphasizing the importance of dissemination and sharing of health information [7,12].
Comment 11) '...significant changes in HDL-C and TG' - you didn't measure change scores but used a multiple linear regression that was adjusted for almost all variables to assess the relationship between perception of body shape and HDL-C and TG. The way this is currently written suggests a causative as opposed to a predictive relationship, please amend.
<Response 11>
Thank you for your comments. We have corrected the manuscript accordingly.
[Page 15,line 250-255]
The results of regression analysis after adjusting for almost all variables, revealed that the group of participants that recognized their body image correctly showed high HDL-C and low TG levels when they actively used nutrition labeling, but the other group of participants that did not correctly recognize their body size and overestimated or underestimated their body shape did not show significant changes in HDL-C and TG levels even if they actively use nutrition labeling.
Comment 12) 'although most health information is overwhelming' - this is a strong generalisation. It may be that most health information is perceived as overwhelming because it is either poorly
communicated or is not tailored to be meaningful at an individual level - the latter further supports your introductory comments
<Response 12>
Thank you for your comments. We agreed with your advice. We deleted the phrase.
Round 2
Reviewer 2 Report
Thank you for your revisions. Please, take a look at the formatting of tables.
Author Response
Thank you for the comments regarding our manuscript. It is our pleasure to submit a new version of our paper that takes the reviewers’ critiques into account. We carefully considered and addressed each point raised by the reviewers. Below, reviewer comment is followed by the authors’ response. Modifications made to the manuscript are indicated by yellow highlighting.
Thank you very much for your interest and careful review of our study.
Responses to the review by reviewer #2:
Comment 1) Thank you for your revisions. Please, take a look at the formatting of tables.
<Answer>
Thank you for your comment. We amended formatting of tables..

Reviewer 3 Report
Thank you for the opportunity to review a revised version of this manuscript, it's pleasing to see that the authors have made considerable changes in response to reviewers' comments.
I have a few more minor recommendations before I am happy to recommend acceptance. Please consider addressing these matters.
Introduction - the introduction is much improved, but please consider how you transition between the final three paragraphs of this section (lines 47-73; mostly highlighted text) as these form the central rationale for the study. Linking these paragraphs, will lead the reader to view the study as a more natural line of inquiry.
Thank you for the addition of Figure 1, and the further additions to the methods section.
Lines 107 - 111 - are more number bullets required here, as you list 4/5 categories, but only provide numbers for two?
Line 120 - 121 - have you a reference for these exercise thresholds? As it may be argued that 150min of moderate intensity exercise, whilst 10% longer than 135min of high intensity exercise, is likely >10% easier than its high intensity equivalent and so this adjustment based solely on time is a flawed comparison
128 - I agree with the comment, but reference(s) are required to support the assertion regarding BMI and East Asian populations
Table 2 - a different caption may be required, as this is the same as Table 1 at present
166 - 178 - please amend formatting for this section
190 - please amend the word correct to healthy, as per Table 5
I would like the authors to consider the issue of over-adjustment. My understanding is that given Y = β0 + β1Χ1 + εi i.e. Y = intercept + slope x independent variable + error, by adjusting for a large number of variables across a large cohort, one is more likely to find significant predictors of dependent variables as an increasing number of independent variables are included in the model. Thus, whilst statistically significant, these values may be of limited practical difference, hence low β values with concomitantly low/highly significant p values.
The authors may also need to consider the collinearity of these variables, especially as they may pertain to the values presented in Tables 3 and 4, and thus the goodness of fit of the multiple linear regression e.g. BMI and waist circumference are likely highly correlated with each other, and other explanatory variables.
I apologise for not picking up these issues sooner, but they became clearer with improved writing and data presentation. This isn't a bad thing, as the transparency (and thus hopefully rigour) of the work is much improved.
Author Response
Thank you for the comments regarding our manuscript. It is our pleasure to submit a new version of our paper that takes the reviewers’ critiques into account. We carefully considered and addressed each point raised by the reviewers. Below, reviewer comment is followed by the authors’ response. Modifications made to the manuscript are indicated by yellow highlighting.
Thank you very much for your interest and careful review of our study.
※ Please see the attachment
Dear Editor:
Thank you for the comments regarding our manuscript. It is our pleasure to submit a new version of our paper that takes the reviewers’ critiques into account. We carefully considered and addressed each point raised by the reviewers. Below, reviewer comment is followed by the authors’ response. Modifications made to the manuscript are indicated by yellow highlighting.
Thank you very much for your interest and careful review of our study.
Responses to the review by reviewer #3:
Comment 1) Introduction - the introduction is much improved, but please consider how you transition between the final three paragraphs of this section (lines 47-73; mostly highlighted text) as these form the central rationale for the study. Linking these paragraphs, will lead the reader to view the study as a more natural line of inquiry.
<Answer>
Thank you for your comment. We have modified the sentences accordingly.
[Page 2, line 47-67]
To practice a healthy lifestyle, it is important to cultivate habits that increase health-related knowledge and beliefs, improve attitude, and enhance self-efficacy [14]. Tailoring, which can be defined as any of the several methods applied for creating communications that are customized, is generally considered to be a useful method of increasing the effectiveness of health interventions, through the delivery of tailored print, as well as telemedicine or eHealth applications [15,16]. Body image could be an important personal factor for tailoring. Development of a healthy perception of one’s body image is the first step in preventing and managing obesity and is essential for forming healthy eating habits as well as for healthy weight control [17,18]. Body image refers to how one perceives, feels, and thinks about one’s body [19]. Distorted body images cause psychological problems such as over-exercising, eating disorders, decline in self-esteem, depression, and physical problems such as malnutrition, osteoporosis, digestive problems, and cardiovascular diseases [20,21]. Previous studies have focused on how intervention strategies affect body image but little attention has been paid towards research using body image as a resource to develop intervention strategies for healthy eating habits [22,23].
Comment 2) Thank you for the addition of Figure 1, and the further additions to the methods section.
<Answer>
Thank you for your comment. We added to the methods section.
[Page 3, line 92-95]
Between 2013 and 2018 a total of 47,217 participants were enrolled. Respondents who aged less than 19 were excluded from these analyses (n=6,453). We also excluded respondents diagnosed with hypertension, diabetes, or dyslipidemia from the study (n=8,065/ n=3,163/ n=5,621), and those with missing values for the diagnosis of the diseases (n=558).
Comment 3) Lines 107 - 111 - are more number bullets required here, as you list 4/5 categories, but only provide numbers for two?
<Answer>
Thank you for your feedback. The purpose of this study was to analyze the effects of active use of nutrition labeling. Thus we categorized the variable with two parts: actively used and not actively used.
Comment 4) Line 120 - 121 - have you a reference for these exercise thresholds? As it may be argued that 150min of moderate intensity exercise, whilst 10% longer than 135min of high intensity exercise, is likely >10% easier than its high intensity equivalent and so this adjustment based solely on time is a flawed comparison
<Answer>
Thank you for your comment. We are sorry that there was an error in writing sentences. We did not use two criteria and applied only one criterion. The sentence was modified as follows.
[Page 4, line 126-128]
Aerobic exercise habits were based on the amount of aerobic exercise per week, with 150 min of exercise as the cutoff
Comment 5) 128 - I agree with the comment, but reference(s) are required to support the assertion regarding BMI and East Asian populations
<Answer>
Thank you for your comment. We presented the evidence for the obesity criteria in Korea in reference literature 32, and deleted the 127-128th line sentence because it could cause confusion.
[Page 4, line 132-135]
BMI was defined as underweight, normal, and obese based on obesity criteria in South Korea (<18.5, 18.5–25, and >25, respectively) [32].
Comment 6) Table 2 - a different caption may be required, as this is the same as Table 1 at present
<Answer>
Thank you for your comments. We corrected captions
[Page 4, line 164]
Table 1. General characteristics of the study population: categorical variables
[Page 6, line 171]
Table 2. General characteristics of the study population: continuous variables.
Comment 7) 166 - 178 - please amend formatting for this section
<Answer>
Thank you for your comments. We amended formatting for the section.
Comment 8) 190 - please amend the word correct to healthy, as per Table 5
<Answer>
Thank you for your comments. We corrected the word
[Page 10, line 200–202]
In contrast, in the group that comprised participants who did not have healthy awareness of their body size, there were not significant differences in HDL-C and TG levels according to the active use of nutrition labeling (β = 0.008, p = 0.3758; β = -0.028, p = 0.1754; β = 0.006, p = 0.6303; β -0.028, p = 0.3503)
Comment 9) I would like the authors to consider the issue of over-adjustment. My understanding is that given Y = β0 + β1Χ1 + εi i.e. Y = intercept + slope x independent variable + error, by adjusting for a large number of variables across a large cohort, one is more likely to find significant predictors of dependent variables as an increasing number of independent variables are included in the model. Thus, whilst statistically significant, these values may be of limited practical difference, hence low β values with concomitantly low/highly significant p values.
The authors may also need to consider the collinearity of these variables, especially as they may pertain to the values presented in Tables 3 and 4, and thus the goodness of fit of the multiple linear regression e.g. BMI and waist circumference are likely highly correlated with each other, and other explanatory variables.
<Answer>
Thank you for your comments. Regarding your comment, we test for multicollinearity between independent variables as follows. By the criteria of VIF, there were slight collinearity between variables. So, we decided to exclude similar variables based on your recommendation (waist circumference and Proportion of carbohydrate among total energy intake).
|
Variables |
VIF |
|
Active use of nutrition labelling |
1.1131 |
|
Sex |
2.2117 |
|
Age (years) |
1.9987 |
|
Educational level |
1.3277 |
|
Economic status |
1.1962 |
|
Household income |
1.0879 |
|
Marital status |
1.5852 |
|
Residence Area |
1.0812 |
|
Perceived health status |
1.0671 |
|
Stress awareness |
1.0701 |
|
Alcohol intake |
1.1690 |
|
Smoking status |
1.5799 |
|
Aerobic exercise habits |
1.2077 |
|
Walking for more than 10 minutes |
1.2150 |
|
Frequency of eating out |
1.3255 |
|
Family history for hyperlipidemia |
1.0184 |
|
BMI(kg/m2) |
4.9729 |
|
Waist circumference (cm) |
6.0063 |
|
Cholesterol (mg) |
1.1379 |
|
Proportion of carbohydrate among total energy intake (%) |
1.3286 |
|
Total energy intake (Kcal) |
1.3349 |
|
Year |
1.1123 |
After change the model which excluded two variables, multicollinearity issues were disappeared, and overall results were not changed with previous version, except to slight magnitude changes of coefficient. Considering re-analysis, we revised the overall manuscript and table.
|
Variables |
VIF |
|
Active use of nutrition labelling |
1.1111 |
|
Sex |
1.8564 |
|
Age (years) |
1.8967 |
|
Educational level |
1.3254 |
|
Economic status |
1.1921 |
|
Household income |
1.0868 |
|
Marital status |
1.5747 |
|
Residence Area |
1.0808 |
|
Perceived health status |
1.0659 |
|
Stress awareness |
1.0699 |
|
Alcohol intake |
1.1247 |
|
Smoking status |
1.5702 |
|
Aerobic exercise habits |
1.2051 |
|
Walking for more than 10 minutes |
1.2149 |
|
Frequency of eating out |
1.3224 |
|
Family history for hyperlipidemia |
1.0184 |
|
BMI(kg/m2) |
1.1313 |
|
Cholesterol (mg) |
1.1331 |
|
Total energy intake (Kcal) |
1.1919 |
|
Year |
1.1023 |
